# Isolation and Characterization of a Cold-Adapted Bacteriophage for Biocontrol of *Vibrio parahaemolyticus* in Seafood

**DOI:** 10.3390/foods14152660

**Published:** 2025-07-29

**Authors:** Zhixiang Nie, Xiangyu Cheng, Shengshi Jiang, Zhibin Zhang, Diwei Zhang, Hanfang Chen, Na Ling, Yingwang Ye

**Affiliations:** 1School of Food and Biological Engineering, Key Laboratory for Animal Food Green Manufacturing and Resource Mining of Anhui Province, Engineering Research Center of Bio-Process, Ministry of Education, Hefei University of Technology, Hefei 230009, China; 2023171462@mail.hfut.edu.cn (Z.N.); 15738298827@163.com (X.C.); 2023171405@mail.hfut.edu.cn (S.J.); 13035081150@163.com (Z.Z.); zhangdiwei2021@163.com (D.Z.); 2 Guangdong Provincial Key Laboratory of Microbial Safety and Health, State Key Laboratory of Applied Microbiology Southern China, Institute of Microbiology, Guangdong Academy of Sciences, Guangzhou 510070, China; chenhanfangh@163.com

**Keywords:** *Vibrio parahaemolyticus*, bacteriophage, aquaculture, biocontrol, seafood

## Abstract

*Vibrio parahaemolyticus* (*V. parahaemolyticus*) is a preeminent seafood-borne pathogen, imposing significant economic burdens on global aquaculture. The escalating prevalence of multidrug-resistant strains has accentuated the critical urgency for developing sustainable biocontrol strategies. In this study, a bacteriophage designated vB_VPAP_XY75 (XY75) was isolated and biologically characterized to establish an effective control against *V*. *parahaemolyticus*. XY75 exhibited remarkable specificity toward *V. parahaemolyticus*, effectively lysing 46.2% of the target strains while showing no lytic activity against non-target bacterial species. Morphological characterization confirmed its taxonomic assignment to the *Myoviridae* family, featuring an icosahedral head (40 ± 2 nm) and contractile tail (60 ± 2 nm). XY75 demonstrated strong environmental tolerance, remaining stable at pH 4–11 and temperatures as high as 50 °C. At an optimal multiplicity of infection (MOI = 0.01), XY75 achieved a peak titer of 8.1 × 10^10^ PFU/mL, a 5 min latent period, and burst size of 118 PFU/cell. Critically, XY75 reduced *V. parahaemolyticus* in salmon by more than 5.98 log CFU/g (99.9%) within 6 h at 4 °C, demonstrating exceptional cold tolerance and lytic activity. Genomic analysis confirmed that no virulence or antibiotic resistance genes were present. These results establish XY75 as a safe and efficacious biocontrol candidate for seafood preservation, with particular utility under refrigerated storage conditions.

## 1. Introduction

*V. parahaemolyticus* is a Gram-negative, non-spore-forming, halophilic bacterium of the genus *Vibrio* [1,2]. *Vibrio parahaemolyticus* is a common food-borne pathogen causing acute gastroenteritis, stomach cramping, and gastrointestinal discomfort, primarily caused by contaminated seafood [3,4]. The Centers for Disease Control and Prevention (CDC) have estimated that the bacterium is responsible for over 45,000 cases of acute gastroenteritis in the United States on an annual basis [5]. Although *V. parahaemolyticus* has historically been thought of as a coastal disease, recent research indicated that it may increasingly colonize freshwater settings inland [6]. Increased demand for seafood in inland areas and long cycles of cold chain transportation might be the causes of this phenomenon. Furthermore, *V. parahaemolyticus* is a prevalent pathogenic bacterium causing inflammation, ulceration, and mortality in aquatic animals [7]. Consequently, safeguards against *V. parahaemolyticus* are essential for maintaining the robust growth of aquaculture as well as for ensuring the safety of seafood.

*V. parahaemolyticus* is primarily characterized by acute clinical symptoms, leading to extensive administration of antibiotics for early treatment. However, the overuse of antibiotics has accelerated the emergence of “super-resistant bacteria” posing a serious threat to human health [8,9]. The widespread use of biocides and antimicrobial preservatives in agriculture and food processing may impose selective pressure on bacteria at sublethal concentrations. This may drive the emergence of broad-spectrum resistance mechanisms, or co-select antibiotic resistance genes through shared mobile genetic elements, ultimately conferring cross-resistance to clinically relevant antibiotics [10]. Studies confirm multidrug resistance in *V. parahaemolyticus* against antibiotics including ampicillin, streptomycin, amikacin, penicillin, and ceftazidime [11]. Additionally, the persistent accumulation of antibiotic residues in water environments exacerbates ecological risks by facilitating antibiotic resistance dissemination, with potential indirect implications on food safety through environmental contamination pathways [12].

Therefore, developing safe, efficient, and side-effect-free antibiotic alternatives is imperative. Compared to chemical agents, biocontrol offers an efficient strategy for suppressing pathogenic organisms. Bacteriophages, as kinds of environmentally friendly biocontrol agents that are abundant on the Earth, have broad prospects in the prevention and control of bacterial infections [13,14]. Phage therapy utilizes the bactericidal activity of bacteriophages to inhibit or eliminate pathogenic bacteria [15]. Specifically, bacteriophages offer distinct advantages over conventional chemical approaches, exhibiting high safety and specificity, minimally disrupting normal flora, and negligibly impacting food nutrients and flavor [16]. Recently, the U.S. Food and Drug Administration (FDA) has approved several commercial phage preparations, targeting *Escherichia coli*, *Listeria*, and *Salmonella*, which have demonstrated efficacy in controlling the contamination of common food-borne pathogens in food processing [17,18,19]. Currently, research on phages targeting *V. parahaemolyticus* remains limited, with a paucity of mature application-oriented studies.

In this study, we isolated phage XY75, with broad-spectrum and specific lytic activity against *V. parahaemolyticus*. Environmental tolerance, genomic characteristics, and inhibitory efficacy in salmon of phage XY75 were systematically analyzed. The results demonstrated that phage XY75 exhibited promising potential for mitigating *V. parahaemolyticus* contamination and suppressing its proliferation in salmon, thereby offering a novel biocontrol strategy for food-borne pathogen management in the safety of seafood and aquaculture.

## 2. Materials and Methods

### 2.1. Bacterial Strains and Growth Conditions

This study utilized a total of 26 *V. parahaemolyticus* strains and 8 strains of other non-Vibrio bacteria provided by Hefei University of Technology, China, which were stored at −80 °C in 50% (*v*/*v*) glycerol (Table 1). All *V. parahaemolyticus* strains were cultured in 5 mL of tryptic soy broth (TSB, HuanKai Microbial, Guangzhou, China) and incubated at 37 °C for 12 h.

### 2.2. Isolation, Purification, and Propagation of Bacteriophages

The isolation of phage XY75 was conducted from sewage samples collected from an effluent treatment plant in Hefei, China as described previously, with minor modifications [20]. Most bacteria were removed by vacuum filtration of sewage through a 0.45 µm filter membrane (Jin Jing, Shanghai, China) using a recirculating water vacuum pump (Great Wall, Zhengzhou, China); subsequently, the mixture was vacuum-filtered through a 0.22 µm filter membrane (Jin Jing, Shanghai, China) to collect phage particles, which were then eluted via ultrasonication. The suspension was subjected to filtration through a 0.45 µm filter (Jin Teng, Tianjin, China). An equal volume of Luria–Bertani (LB) broth (Haibo Microbial, Qingdao, China) supplemented with 2 mM CaCl_2_ was added to the phage suspension, followed by inoculation with 50 µL of the bacterial strain described in Section 2.1 of the log phase. The culture was then incubated with shaking at 37 °C for 8 h. After centrifugation (4000× *g*, 5 min), the supernatant was collected for phage titer determination using the double-layer agar method (the bottom layer was 1.5% TSA (HuanKai Microbial, Guangzhou, China) solid medium and the upper layer was 0.4% TSB soft agar layer containing bacteria). A single translucent phage plaque was picked, resuspended in SM buffer (50 mM Tris-HCl, 100 mM NaCl, 10 mM MgSO_4_, pH 7.5), and mixed thoroughly. Homogeneous plaque morphology was observed following 3–4 rounds of purification using the double-layer agar technique.

### 2.3. Transmission Electron Microscopy (TEM)

Phage morphology was analyzed by TEM using a modified protocol [21]. Briefly, purified phage particles were adsorbed onto carbon-coated copper grids for 4 min and then excess suspension was removed by filter paper adsorption. Copper grids were stained with 2% (*w*/*v*) phosphotungstic acid for 3 min, blotted to remove residual stain, and air-dried. The morphology of the phage was observed by a field-emission TEM (JEM-2100F, JEOL Ltd., Tokyo, Japan), at an acceleration voltage of 200 kV.

### 2.4. Phage Lytic Range

The host range was assessed based on their ability to form plaques on bacterial lawns using the double-layer plate spreading method, based on the clarity of the zone of inhibition [22]. Then, 100 μL of different strains in log phase were mixed with 10 mL of TSB (0.4% agar) and poured onto plates containing 1.5% of TSA bottom layer. A volume of 10 uL of phage suspension was dropped onto the coagulated solid medium. After that, the plates were incubated at 37 °C for 3 h and observed.

### 2.5. Multiplicity of Infection (MOI)

To determine the optimal MOI, *V. parahaemolyticus* strain GD75 was grown to logarithmic phase and diluted to 10^8^ CFU/mL, and then mixed with phage based on ratios of MOIs of 1000, 100, 10, 1, 0.1, 0.01, and 0.001 [23]. The mixture was incubated at 37 °C and 200 rpm for 4 h, and the supernatant was collected by centrifugation for phage titer determination. The experiment was repeated three times.

### 2.6. One-Step Growth Assay

The one-step growth of XY75 was determined based on previous reports, with some modifications [24]. Bacterial cultures were grown to a pre-logarithmic phase at a concentration of 10^8^ CFU/mL. Host bacteria were mixed with phage suspension at an optimal MOI of 1:100 and preincubated for 2 min at 37 °C. The mixture was then centrifuged at 10,000 rpm/min for 1 min, and the pellet was collected. The pellet was washed twice and resuspended in TSB medium preheated at 37 °C and incubated at 37 °C and 200 rpm. Samples were collected at 0 min, at 5 min intervals for the first 20 min, and at 10 min intervals thereafter. Each sample was centrifuged at 12,000 rpm for 30 s. The potency was determined at each time point by extraction with a 0.45 µm filter tip. A one-step growth curve was plotted with infection time as the horizontal coordinate and phage potency as the vertical coordinate.

### 2.7. Stability Tests of Bacteriophage XY75

A slight modification of the previous method [25] was used to test the stability of XY75. To assess pH stability, HCl and NaOH were used to adjust the different pHs (3, 4, 5, 6, 7, 8, 9, 10, 11, and 12) of TSB. Phages were equally added to this TSB and placed for 1 h at 37 °C, followed by determination of phage titers. To assess thermal stability, phages (10^9^ PFU/mL) were exposed to various temperatures (25 °C, 30 °C, 37 °C, 40 °C, 50 °C, 60 °C, 65 °C, 70 °C, 75 °C, and 80 °C) for 1 h.

### 2.8. Extraction of DNA

The Sodium Dodecyl Sulfate (SDS)-Proteinase K protocol was used to extract the genomic DNA of phage XY75 [26]. Briefly, the phage was precipitated overnight with 15% (*w*/*v*) PEG 8000 and 0.5 M NaCl at 4 °C. Subsequently, the phage was centrifuged at 12,000× *g* for 20 min and then resuspended in SM buffer. The concentrated phage particles were treated with 1% (*w*/*v*) SDS and 100 μg/mL proteinase K and incubated at 56 °C for 2 h. Thereafter, an equal volume of phenol–chloroform–isoamyl alcohol (25:24:1) was added. The aqueous upper phase was collected and was precipitated with 2 volumes of ethanol and 0.1 volumes of 3 M sodium acetate (pH 5.2) at −20 °C overnight. Finally, the DNA pellet was collected by centrifugation (12,000 rpm, 15 min), washed twice with 70% ethanol, and resuspended in TE buffer (10 mM Tris-HCl, 1 mM EDTA, pH 8.0) for storage at −20 °C.

### 2.9. Genome-Wide Analysis and Phylogenetic Analysis

The extracted DNA was sequenced using the Illumina HiSeq X Ten platform (Illumina, San Diego, CA, USA). Reads were QCed by Cutadapt 1.9.1, then assembled using Velvet 1.2.10, and gaps were filled with SSPACE 3.0 and GapFiller 1–10. Whole genome sequences were compared using the National Center for Biotechnology Information (NCBI) database (https://blast.ncbi.nlm.nih.gov/Blast.cgi, accessed on 18 November 2024) [27]. A heatmap of the genome sequences was constructed of the 19 phage genomes with the highest sequence similarity to the XY75 phage using the Virus Intergenomic Distance Calculator (VIRIDIC) online website (https://rhea.icbm.uni-oldenburg.de/VIRIDIC/, accessed on 18 November 2024) [28]. Whole genome sequences of phages were compared to host virulence genes in the Virulence Factor Database (https://www.mgc.ac.cn/VFs/, accessed on 18 November 2024) using the Basic Local Alignment Search Tool for Nucleotide (BLASTn) (https://blast.ncbi.nlm.nih.gov/Blast.cgi, accessed on 18 November 2024) (coverage ≥ 95%, identity ≥ 90%) to determine whether phages carry virulence genes [20]. Manual annotation was performed using XY75 proteins predicted with the NCBI Conserved Domain Database (NCBI CDD) Hpred tool. The Circular Genome Viewer (CGView) (v1.5.0) [29] (https://www.bioinformatics.org/cgview/gallery.html, accessed on 18 November 2024) was used to generate XY75 genome maps. Whole genome sequences of phages were visualized using EasyFigure (2.2.4). A phylogenetic tree of RNA polymerase was constructed using MEGA X by the neighbor-joining method and 1000 bootstrap replicates. The Newick format was used in the creation of a phylogenetic tree by iTOL (v6.8.1) (https://itol.embl.de).

### 2.10. Prevention and Control Applications for Salmon

Fresh-cut salmon was purchased from a local fish market. The fish was rinsed twice with sterilized physiological saline, aseptically cut into equal-weight portions (10 g), and surface-decontaminated using a sterilizing UV lamp for 2 h. Following this treatment, no detectable bacteria were recovered from the samples. The samples were then artificially spiked with *V. parahaemolyticus* GD80 (1 × 10^6^ CFU/g fish fillet) and dried under laminar airflow for 10 min to simulate cross-contamination likely encountered during seafood processing. Subsequently, 100 μL of SM buffer containing bacteriophage XY75 (at MOIs of 10, 1, 0.1, and 0.01) was added dropwise onto the contaminated fish portions. The resulting samples were incubated at 4 °C and 37 °C for 6 h. Every 2 h, 10 mL of normal saline (0.9% NaCl, sterile) was added to 1 g of sample to make the dilutions. Each sample was homogenized for 2 min using a stomacher, and viable *V. parahaemolyticus* cells were enumerated by plating the homogenate onto TCBS agar.

## 3. Results

### 3.1. Isolation and Morphological Analysis of Phage XY75

Phage XY75 was isolated from Hefei sewage using strain GD75 as host. It formed transparent plaques (0.5–1 mm diameter; Figure 1A). TEM analysis (Figure 1B) revealed XY75 belongs to the *Myoviridae* family, with an icosahedral head (40 ± 2 nm nm diameter) and a contractile tail (60 ± 2 nm length).

### 3.2. Host Range of XY75

In this study, the host range of XY75 was determined using 26 *V. parahaemolyticus* strains and 8 strains representing other bacterial species (Table 1). The complete resistance of non-*V. parahaemolyticus* strains to XY75 infection underscored its high specificity for *V. parahaemolyticus*. Notably, 12 of the 26 *V. parahaemolyticus* strains were susceptible to XY75, indicating a relatively broad host range within the species. These findings highlight substantial potential of XY75 as a targeted bactericidal agent against *V. parahaemolyticus*.

### 3.3. Biological Characteristics of XY75

The ratio of the number of phages to the number of indicator cells during infection is called the infection complex. The optimal multiplicity of infection is the one that yields the highest titer. At an optimal multiplicity of infection (MOI = 0.01), XY75 achieved a peak titer of 8.1 × 10^10^ PFU/mL (Figure 2A). The optimal MOI for phage XY75 was favorable for practical applications. The one-step growth curve of phage XY75 showed (Figure 2B) that 0–5 min was the adsorption period of phage XY75, 10–20 min was the lysis period of phage XY75, and 30–120 min was the plateau period of phage XY75. There was no obvious change in the activity of phage XY75 below 50 °C; the phage activity was severely decreased after 1 h of treatment at 60–70 °C; and the phage was completely inactivated after 1 h of treatment at 80 °C (Figure 2C). Phage XY75 exhibited remarkable resistance to acid–base fluctuations and thermal stress. However, exposure to extreme pH conditions (strong acid/alkali) or high temperatures led to disruption of its protein spatial conformation, specifically impairing the receptor-binding proteins and abrogating bacterial lytic activity. Phage XY75 was completely inactivated for pH ≤ 3 or pH ≥ 12 (Figure 2D).

### 3.4. Genome Sequence Analysis

Illumina sequencing determined the double-stranded DNA (dsDNA) genome of bacteriophage XY75 to be 44,860 bp in length, with a GC content of 48.68%. The results of BLASTn comparison showed that the similarity between phage XY75 and *Vibrio* phage vB_VpaP_MGD1 (MT501516.1) was 100% (96.03% coverage); the similarity between phage XY75 and *Vibrio* phage vB_VpaP_1701 (ON872379) was 98% (95.28% coverage). The VIRIDIC heatmap (Figure 3) showed that phage XY75 exhibited significant intergenomic similarity with other *vibrio* phages. Furthermore, no antibiotic resistance genes (ARGs) or virulence-associated genes (VAGs) were detected in the genome of phage XY75. The morphology of phage XY75 particles and functional analysis of its predicted gene products suggested that it is a characteristic member of the *Myoviridae* family. Possessing its own RNA polymerase (ORF 29), ICTV classification places this phage within the *Autographiviridae* subfamily (Figure 4). Consequently, the genomic sequences of 43 *Autographivirina Vibrio* phages were downloaded from the ICTV database for the purpose of phylogenetic-tree analysis. Phylogenetic-tree analysis revealed that XY75 clustered within the branch of the *Maculvirus* genus, consistent with the whole-genome similarity result.

As demonstrated in Figure 5, the genome of phage XY75 is 44,860 bp in length, with a GC content of 48.68%. Genomic annotation analysis predicted 50 open reading frames (ORFs) without tRNA. Of these, 27 ORFs were annotated as functional proteins, and the rest were annotated as hypothetical protein. The functional proteins of XY75 were divided into six modules, including DNA metabolism; lysis; packaging; structure; additional function; and the hypothetical proteins module. In order to interpret the relationships of the phages at the genomic level, the complete genome sequence of XY75 was compared, vB_VpaP_MGD1 and *Vibrio* phage vB_VpP_3 (PP834370). As shown in Table 2, the phage genome carries multiple tail proteins, including tail tube protein (ORF 36, ORF 37), tail protein (ORF 38), and tail fiber protein (ORF 42). Moreover, it also contains lytic enzymes (ORF 41, ORF 49), bacterial Ig-like domain family protein (ORF 48), and other functional proteins. As shown in Figure 6, XY75 was homologous to vB_VpaP_MGD1 and vB_VpP_3 in a very large part of the genome, and the nucleotide identities were both not less than 92%. It has been suggested that phages exhibiting more than 40% protein similarity are likely to belong to a specific genus. Based on this method, we determined that phage XY75 belongs to the same genus as vB_VpaP_MGD1 and vB_VpP_3.

### 3.5. Food Applications of Phage XY75

Salmon contaminated with *Vibrio parahaemolyticus* were treated with phage XY75 at different MOIs. After being stored at 37 °C for 6 h, the survival numbers of *Vibrio parahaemolyticus* increased by 1.78 log CFU/mL (MOI = 10), 1.87 log CFU/mL (MOI = 1), 2.2 log CFU/mL (MOI = 0.1), and 2.14 log CFU/mL (MOI = 0.01) compared to the control group. However, when treated at 4 °C for 6 h, the survival numbers of *Vibrio parahaemolyticus* reduced by 5.98 log CFU/mL (MOI = 10), 2.01 log CFU/mL (MOI = 1), and 1.10 log CFU/mL (MOI = 0.01), while at MOI = 0.01, the survival numbers increased by 0.08 log CFU/mL (Figure 7A,B). The experiment demonstrated that high-titer phages (10^7^ PFU/mL, MOI = 10) exhibited excellent bactericidal performance at both 37 °C and 4 °C. Notably, the 4 °C group achieved a 99.9% bactericidal rate after 6 h.

## 4. Discussion

One of the main causes of illnesses linked to seafood is the bacterium *V. parahaemolyticus*, which affects a variety of aquatic creatures, including fish, shellfish, and salmon [30,31]. This bacterium has the ability to develop biofilms in food processing environments, posing a significant threat to food safety within the seafood and aquaculture sectors [32]. Antibiotic-resistant *V. parahaemolyticus* strains have become more common in recent years [33,34], highlighting the urgent need for effective antimicrobial solutions. Lytic bacteriophages utilize the cellular machinery of host bacteria for replication, causing the bacteria to lyse and release the phage progeny into the surrounding environment, and this capability offers a promising approach for combating bacterial infections [35,36,37]. In this study, bacteriophage XY75, which specifically lyses *V. parahaemolyticus,* was isolated and biologically characterized to establish an effective control against *V*. *parahaemolyticus*.

Morphological analysis revealed an icosahedral head with a diameter of 40 ± 2 nm and a contractile tail measuring 60 ± 2 nm in length. These characteristics are typical of *Myoviridae* family members and suggest effective host infectivity. Host range assays confirmed a high degree of specificity for *V. parahaemolyticus*, which showed that the phage lysed 12 out of the 26 strains examined (46.2% coverage). The host range of the phage was determined principally by phage tail fibronectin [38]. Research findings have demonstrated the successful transformation of the fibronectin-associated cell-to-cell adhesion zone in Mycoplasma T3, which has enabled the expansion of its host range [39]. The one-step growth curve of our phage XY75 exhibited a peak titer of 8.1 × 10^10^ PFU/mL at an MOI of 0.01, a short latent period of 5 min, and robust lytic capacity, facilitating future industrial production of phage formulations. Seafood processing environments are inherently complex and variable, necessitating a robust environmental tolerance for phages to sustain their bioactivity. The shelf-life of salmon at 2 °C was determined to be approximately 10 days [40]. The XY75 phage exhibited excellent tolerance to temperature (20–50 °C) and pH (4–11). Therefore, compared to other phages, XY75 exhibits relatively broad temperature and pH stability, collectively supporting its capability for rapid bacterial eradication.

Illumina sequencing identified a 44,860 bp dsDNA genome with a GC content of 48.68%. VIRIDIC and BLASTN analyses showed 100% nucleotide similarity (96.03% coverage) with *Vibrio* phage vB_VpaP_MGD1 and more than 92% homology with conserved regions of vB_VpP_3. Phages XY75 and H256D1 were both located in the same evolutionary branch [41]. Among 50 predicted ORFs, 26 were functionally annotated, including DNA metabolism, structural assembly, packaging, and lysis modules. Practical application of this phage ultimately hinges on its safety profile and antimicrobial efficacy. Genomic analysis confirmed XY75 carries no genes associated with antibiotic resistance, toxins, lysogeny, or virulence factors, indicating that this bacteriophage may be harmless to humans and could be used in the food industry. Identification of an RNA polymerase (ORF 29) classified XY75 of the genus *Maculvirus* within the *Autographiviridae* subfamily. The most common structures for tail phages to recognize bacterial receptors are the tail spikes, tail fibers, and tail membrane penetrating proteins [42]. Among these, the tail fibers mediate adsorption by binding to receptors, including LPS, flagella, type 4 pili, and outer membrane porins (ompC, ompF) [43]. The phage genome encodes multiple tail proteins, including tail tube protein A/B (TTPA/TTPB, ORF36/37), tail protein (ORF 38), and tail fiber protein (ORF 42), which may serve as ligands to recognize conserved *Vibrio* receptors, thereby mediating phage adsorption and facilitating efficient recognition of host receptors. However, the fact that bacteriophages rely on receptor recognition to achieve infection implies that resistance to bacteriophages can arise through receptor mutations [39]. Critical lytic enzymes include an endolysin (ORF 49) and a peptidoglycan lytic exotransglycosylase (ORF 41), essential for adsorption, invasion, lysis, and biofilm disruption. An immunoglobulin-like domain protein (ORF 48) might facilitate mucosal surface binding for cell attachment or pathogen clearance [21,44,45].

The popularity of ready-to-eat (RTE) raw fish products, such as sashimi and sushi, has grown significantly in Asia, the Americas, and Europe [46,47,48,49]. However, despite stringent requirements for cold chain maintenance and hygiene during processing, distribution, and retail, these foods present a high risk of microbial contamination. This risk is exemplified by the relatively short shelf life of vacuum-packed salmon stored at 5 °C, which is typically only about 2 weeks [50]. Therefore, RTE raw fish slices serve as an appropriate model to investigate the potential of phage XY57 for the biocontrol of pathogenic *Vibrio* in RTE seafood. In this study, the degree and rate of pathogen reduction is significantly higher when samples were maintained at 37 °C, mimicking improper storage or serving conditions for RTE seafood. When XY75 was applied to pathogen-spiked samples at an MOI of 10, the *V. parahaemolyticus* count decreased by up to 1.0 log unit. The quantity of *V. parahaemolyticus* began to dramatically decline when XY75 was used to treat pathogen-spiked samples (4 °C). In 6 h, this reduction reached up to a 3.0 log reduction from the starting time point, achieving a sterilization rate of 99.9%. Recent studies have demonstrated that, when phages vB_VpaS_1601 and vB_VpaP_1701 were applied at an MOI of 10 to salmon samples, the numbers of *V. parahaemolyticus* were reduced by 1.51 and 1.39 CFU/cm^3^, respectively, with a sterilization rate not exceeding 30% within 6 h [51]. In comparison, phage XY75 exhibited a higher efficacy in eliminating *Vibrio parahaemolyticus* under the same conditions and significantly decreased the bacterial survival rate. Furthermore, phage OMN was applied to oyster samples, resulting in the inactivation of approximately 99% of *V. parahaemolyticus* on the surface of the oyster meat [52]. Our findings are consistent with those reported in these studies. The fast growth of bacteria at 37 °C may hinder the phage–bacteria interaction to some degree. Low-temperature conditions are typically employed in practical environmental applications to inhibit bacterial growth and reproduction, particularly during the shipping of seafood and other items. As a result, phage XY75 has more potential and benefits for use in this sector due to its enhanced bactericidal efficacy at low temperatures. The application of phage XY75 can effectively enhance the safety and quality of products during the transportation and storage of perishable products such as seafood. However, a major challenge for such phage-based biocontrol is the emergence of phage-resistant mutants [53]. The employment of bacteriophage cocktails has been demonstrated to enhance the efficacy of controlling and inactivating the growth of *V. parahaemolyticus* in comparison to the utilization of individual bacteriophage suspensions [54,55]. Therefore, in the future, the application of phage XY75 is expected to be utilized in phage cocktail mixtures as a sterilization system targeting as many *Vibrio parahaemolyticus* as possible.

## 5. Conclusions

Current global climate change has led to an increase in seawater surface temperatures, thereby expanding the geographical distribution and seasonal prevalence of pathogenic *Vibrio* infections. The emergence of antibiotic-resistant *V. parahaemolyticus* underscores the urgent need for sustainable, effective control strategies that can be effectively deployed in food and aquaculture industries. We isolated and characterized the lytic bacteriophage XY75 and evaluated its potential for the prevention and control of *V. parahaemolyticus* in seafood. Phage XY75 is characterized by a short latency and large release, thus facilitating ease of preparation and the subsequent industrial production of phage formulations. In addition, XY75 demonstrates robust functionality across a broad range of temperature (20–50 °C) and pH (4–11), indicating that the phage has a wider range of environmental applications. XY75 could reduce *V. parahaemolyticus* in salmon by more than 5.98 log CFU/g (99.9%) within 6 h at 4 °C, demonstrating superior efficacy to that observed at 37 °C. XY75 demonstrated exceptional cold tolerance and lytic activity, making it an excellent biological control agent for managing *V. parahaemolyticus* in the seafood cold chain. This study provides an innovative strategy and a valuable resource for combating *V. parahaemolyticus* contamination.

## Figures and Tables

**Figure 1 foods-14-02660-f001:**
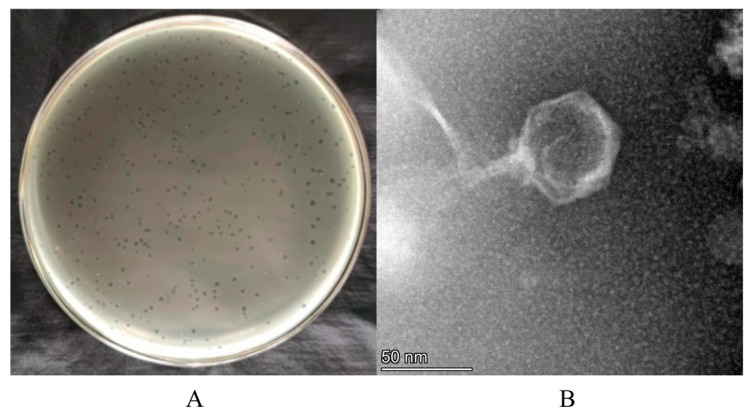
Morphological characterization of phage XY75. (**A**) Phage XY75 spots on TSA agar plates. (**B**) Morphology of phage XY75. Scale bar, 50 nm.

**Figure 2 foods-14-02660-f002:**
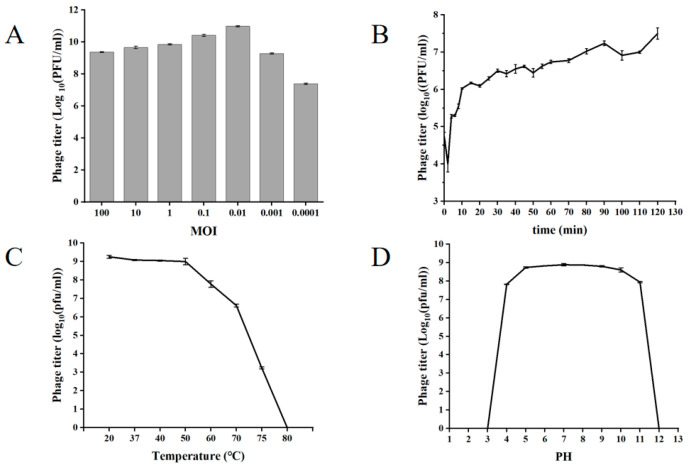
Biological characterization of phage XY75. (**A**) MOI. (**B**) One-step growth curve. (**C**) Temperature stability. (**D**) pH stability.

**Figure 3 foods-14-02660-f003:**
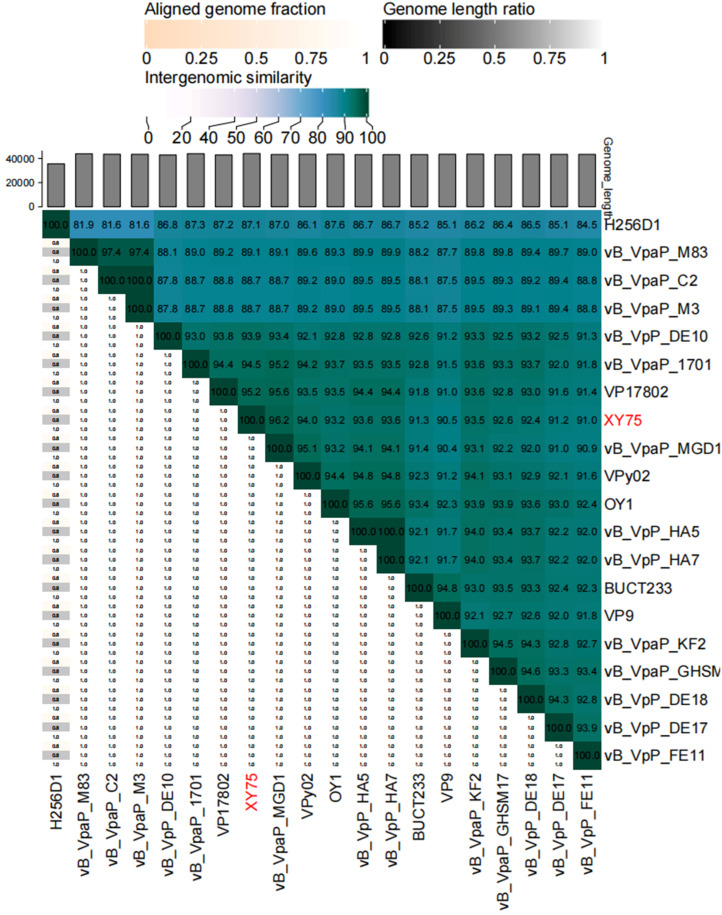
Heatmap of VIRIDIC. The values of the identity percentages range from 0 (white) to 100 (dark green).

**Figure 4 foods-14-02660-f004:**
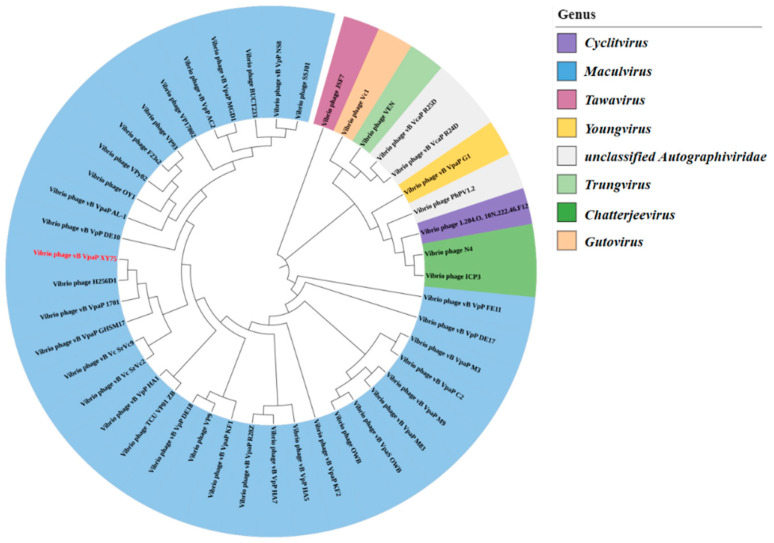
Phylogenetic tree of phage XY75 based on the RNA polymerase sequence.

**Figure 5 foods-14-02660-f005:**
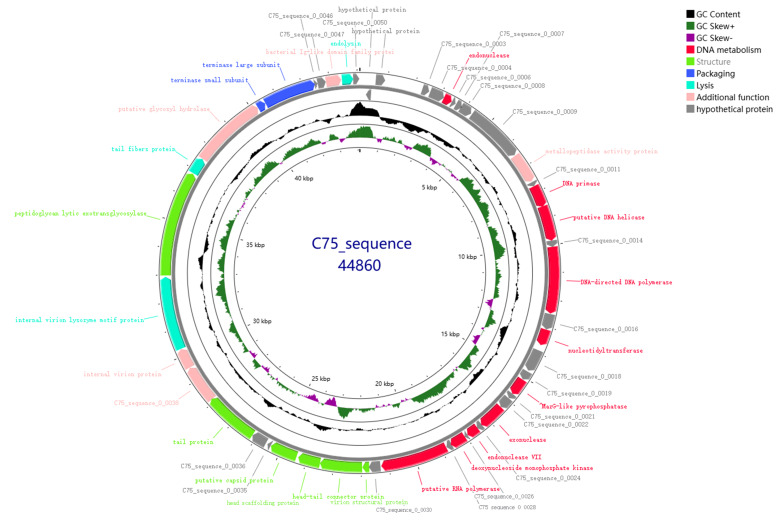
Genome-wide mapping of XY75. Circles from outside to inside: (1) ORFs are indicated in the first and second circles. The direction of the arrow, whether clockwise or counterclockwise, is indicative of the direction of the forward (or reverse) chain coding. For the open reading frames, different colors represent functional categories: hypothetical proteins (grey), DNA metabolism (red), structure (green), packaging (blue), lysis (cyan), and additional function (pink). (2) The third circle is indicative of the GC content of the genome. (3) The fourth circle is indicative of the value of (G − C)/(G + C), which is representative of the GC skew plot that is used to measure the relative contents of G and C (green indicates high GC content; purple indicates low GC content). (4) The innermost circle represents the total length of the genome, expressed in kilobase pairs (Kb).

**Figure 6 foods-14-02660-f006:**
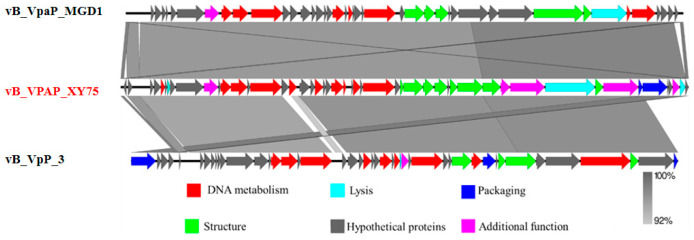
Genome comparison of vB_VPAP_XY75, vB_VpaP_MGD1, vB_VpP_3, and EBPL using Easyfig. The different color arrows represent predicted open reading frames with different functions: hypothetical proteins (grey), DNA metabolism (red), structure (green), packaging (blue), lysis (cyan), and additional function (pink). Shading indicates nucleotide identity between sequences (92–100%).

**Figure 7 foods-14-02660-f007:**
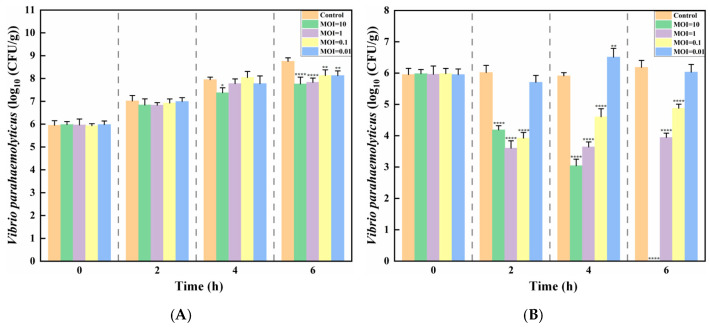
Bacteriostatic effects of phage XY75 on *V. parahaemolyticus* in salmon. (**A**) Salmon, 37 °C; (**B**) salmon, 4 °C. (*) indicates *p* < 0.05, (**) indicates *p* < 0.01, (****) indicates *p* < 0.0001.

**Table 1 foods-14-02660-t001:** Host range of phage XY75.

Strain	Species	Plaque Formation ^a^	Source ^b^
ATCC 17802	*Vibrio parahaemolyticus*	−	ATCC
GD91	*Vibrio parahaemolyticus*	+	Laboratory collection
GD103	*Vibrio parahaemolyticus*	+	Laboratory collection
GD71	*Vibrio parahaemolyticus*	+	Laboratory collection
GD78	*Vibrio parahaemolyticus*	−	Laboratory collection
GD81	*Vibrio parahaemolyticus*	−	Laboratory collection
GD62	*Vibrio parahaemolyticus*	+	Laboratory collection
GD75	*Vibrio parahaemolyticus*	+	Laboratory collection
GD74	*Vibrio parahaemolyticus*	+	Laboratory collection
GD89	*Vibrio parahaemolyticus*	+	Laboratory collection
GD80	*Vibrio parahaemolyticus*	+	Laboratory collection
GD83	*Vibrio parahaemolyticus*	+	Laboratory collection
GD73	*Vibrio parahaemolyticus*	−	Laboratory collection
99-1	*Vibrio parahaemolyticus*	−	Food
66-1	*Vibrio parahaemolyticus*	−	Seawater
124-1	*Vibrio parahaemolyticus*	−	Food
84-1	*Vibrio parahaemolyticus*	+	Food
163-1	*Vibrio parahaemolyticus*	−	Freshwater
240-1	*Vibrio parahaemolyticus*	−	Food
79-1	*Vibrio parahaemolyticus*	−	Food
87-2	*Vibrio parahaemolyticus*	−	Freshwater
84-2	*Vibrio parahaemolyticus*	−	Food
139-1	*Vibrio parahaemolyticus*	−	Food
227-2	*Vibrio parahaemolyticus*	−	Food
51-1	*Vibrio parahaemolyticus*	−	Food
75-2	*Vibrio parahaemolyticus*	−	Food
ATCC 13076	*Salmonella enterica* subsp. *enterica*	−	ATCC
ATCC 19111	*Listeria monocytogenes*	−	ATCC
ATCC 51329	*Cronobacter muytjensii*	−	ATCC
ATCC 19433	*Enterococcus faecalis*	−	ATCC
ATCC 14028	*Salmonella typhimurium*	−	ATCC
ATCC 49128	*Pseudomonas putida*	−	ATCC
ATCC 27583	*Pseudomonas aeruginosa*	−	ATCC
ATCC 6538	*Staphylococcus aureus*	−	ATCC

Note: A total of 34 strains were used as the host to determine the host spectrum of the phage using a spot test. ^a^ Plaque formation; +, clear plaques or slightly turbidity plaques; −, no plaques formed. ^b^ ATCC, American Type Culture Collection.

**Table 2 foods-14-02660-t002:** Gene annotations of phage VB_VPAP_XY75.

ORF	Predicted Function	Best BLAST Hit	Functional Module
5	endonuclease	*Vibrio* phage VP46	DNA metabolism
10	metallopeptidase activity protein	*Vibrio* phage VP46	Additional function
12	DNA primase	*Vibrio* phage vB_VpaP_GHSM17	DNA metabolism
13	putative DNA helicase	*Vibrio* phage vB_VpaP_KF1	DNA metabolism
15	DNA-directed DNA polymerase	*Vibrio* phage vB_VpP_HA1	DNA metabolism
17	nucleotidyltransferase	*Vibrio* phage VP9	DNA metabolism
20	MazG-like pyrophosphatase	*Vibrio* phage F23s2	DNA metabolism
23	exonuclease	*Vibrio* phage vB_VpP_NS8	DNA metabolism
25	endonuclease VII	*Vibrio* phage VP48	DNA metabolism
27	putative deoxynucleoside monophosphate kinase	*Vibrio* phage vB_VpP_DE18	DNA metabolism
29	putative RNA polymerase	*Vibrio* phage H256D1	DNA metabolism
31	virion structural protein	*Vibrio* phage VP48	Structure
32	head–tail connector protein	*Vibrio* phage OY1	Structure
33	head scaffolding protein	*Vibrio* phage vB_VpaP_KF1	Structure
34	putative capsid protein	*Vibrio* phage vB_VpP_FE11	Structure
36	putative tail tubular A	*Vibrio* phage vB_VpaP_KF2	Structure
37	putative tail tubular protein B	*Vibrio* phage VP46	DNA metabolism
38	tail protein	*Vibrio* phage vB_VpP_HA5	Structure
39	internal virion protein	*Vibrio* phage vB_VpP_DE10	Additional function
40	internal virion lysozyme motif protein	*Vibrio* phage vB_VpP_DE10	Additional function
41	peptidoglycan lytic exotransglycosylase	*Vibrio* phage vB_VpP_DE10	Lysis
42	tail fiber protein	*Vibrio* phage VP46	Structure
43	putative glycosyl hydrolase	*Vibrio* phage vB_VpaP_KF1	Additional function
44	terminase small subunit	*Vibrio* phage vB_VpaP_KF1	Additional function
45	terminase large subunit	*Vibrio* phage BUCT233	Additional function
48	bacterial Ig-like domain family protein	*Vibrio* phage vB_VpP_DE10	Additional function
49	endolysin	*Vibrio* phage VP46	Lysis

## Data Availability

The original contributions presented in the study are included in the article, further inquiries can be directed to the corresponding authors.

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
