# Peer review of "Isolation and Characterization of a Cold-Adapted Bacteriophage for Biocontrol of Vibrio parahaemolyticus in Seafood"

_foods, 2025, doi:10.3390/foods14152660_

Round 1
Reviewer 1 Report
Comments and Suggestions for Authors
- Line 45: please include a reference.
- Line 52: “in” is repeated.
- Line 85: the origin/source of phage isolation is not clear.
- Line 91: the concentration of bacterial cells used is not clear.
- Line 95: what is the medium used on the plate?
- Line 96: please provide the meaning of SM.
- Line 123: what was the rotation speed?
- Line 157: 2.10 (?)
- Line 167: which diluent was used? Please describe.
- Figure 1B is very low resolution; it is not possible to clearly distinguish the contractile tail. I strongly suggest selecting another representative image to present.
- Figure 7 is confusing because, if the reduction of the Vibrio population in the sample is being presented, the names of the Y-axes are incorrect.
- Still regarding Figure 7, why is the reduction not presented instead of the absolute count data? Otherwise, it is an unfair comparison between treatments.
- Line 362 and wherever else necessary: please italicize scientific names.
Reviewer 2 Report
Comments and Suggestions for Authors
Dear Authors,
In attachment are some comments in order to improve this Manuscript.
The paper presents an overview about the application of isolated phage XY75 as an antimicrobial agent against the bacteria Vibrio parahaemolyticus which is commonly found in the sea and estuaries, and is responsible for various types of diseases including gastroenteritis. After reading the submitted Manuscript, I consider that some sections can be improved. For this reason, I provide some comments that should be addressed. The following suggestions are presented:
Specific points
Line 51-53 and 55-57: Please, rephrase these sentences for clarity, how the extensive application of fungicides and antimicrobials leads to "potentially fostering cross-resistance to clinically relevant antibiotics" and why only "drug residues in seafood" what about other antimicrobials.
Line 67: Authors should avoid using commercial names, this should be indicated in the Materials section where product used.
Line 80: Which are non-Vibrio bacteria in Table 1 marked as a source of Laboratory collection, please specify?
Line 83: Indicate the manufacturer from whom the TSB was purchased.
Line 91: The full name of the broth is missing, LB. I suppose Luria-Bertani (LB) medium.
Briefly describe methods for 2.7. Stability Tests of Bacteriophage XY75 and 2.8. Extraction of DNA
In general, the discussion should be improved.
The authors should discuss more about the mode of action of phage XY75 against bacteria V. parahaemolyticus and their resistance. Compare the obtained findings with similar assays where other bacteriophages were tested against the same pathogenic bacteria of this manuscript. Highlight the advantages of using phage as an antimicrobial agent and its applicatin. Include future trends to keep working with the obtained data.
Round 2
Reviewer 1 Report
Comments and Suggestions for Authors
The authors answered all questions satisfactorily and updated the representative image of the virus in the new manuscript file.
Reviewer 2 Report
Comments and Suggestions for Authors
Dear Authors,
After a revised Manuscript version 2, the paper could be considered for publication in the journal Foods.